# Solitary Bees Host More Bacteria and Fungi on Their Cuticle than Social Bees

**DOI:** 10.3390/microorganisms11112780

**Published:** 2023-11-16

**Authors:** Markus Thamm, Fabienne Reiß, Leon Sohl, Martin Gabel, Matthias Noll, Ricarda Scheiner

**Affiliations:** 1Behavioral Physiology and Sociobiology, Julius-Maximilians-Universität Würzburg, 97070 Würzburg, Germany; markus.thamm@uni-wuerzburg.de (M.T.); martinsebastian.gabel@llh.hessen.de (M.G.); 2Institute of Bioanalysis, Coburg University of Applied Sciences and Arts, 96450 Coburg, Germany; fabienne.reiss@hs-coburg.de (F.R.); leonsohl97@gmail.com (L.S.); 3Landesbetrieb Landwirtschaft Hessen, Bee Institute Kirchhain, 35274 Kirchhain, Germany; 4Bayreuth Center of Ecology and Environmental Research, University of Bayreuth, 95447 Bayreuth, Germany

**Keywords:** honeybee, red mason bee, cuticular microbiome

## Abstract

Bees come into contact with bacteria and fungi from flowering plants during their foraging trips. The Western honeybee (*Apis mellifera*) shows a pronounced hygienic behavior with social interactions, while the solitary red mason bee (*Osmia bicornis*) lacks a social immune system. Since both visit the same floral resources, it is intriguing to speculate that the body surface of a solitary bee should harbor a more complex microbiome than that of the social honeybee. We compared the cuticular microbiomes of *A. mellifera* (including three European subspecies) and *O. bicornis* for the first time by bacterial 16S rRNA and fungal ITS gene-based high-throughput amplicon sequencing. The cuticular microbiome of the solitary *O. bicornis* was significantly more complex than that of the social *A. mellifera*. The microbiome composition of *A. mellifera* subspecies was very similar. However, we counted significantly different numbers of fungi and a higher diversity in the honeybee subspecies adapted to warmer climates. Our results suggest that the cuticular microbiome of bees is strongly affected by visited plants, lifestyle and adaptation to temperature, which have important implications for the maintenance of the health of bees under conditions of global change.

## 1. Introduction

Honeybees and wild bees are of great ecological and economic importance as pollinators [1]. In the social Western honey bee (*Apis mellifera*), only the fertile queens lay eggs. All of the other tasks are performed by sterile female workers [2]. This globally distributed species shows remarkable intraspecific diversity based on differences in morphology, colony development and behavior [3], leading to 30 subspecies. The three subspecies *Apis mellifera mellifera*, *Apis mellifera carnica* and *Apis mellifera iberiensis* represent typical honeybee subspecies adapted to different climatic conditions. The original distribution of *A.m. mellifera* ranged from the border of the Alps up to Scandinavia and from France to the Ural Mountains [3], i.e., relatively cool areas within Europe. *A.m. carnica* is the dominant honeybee subspecies in central Europe, originally extending to southeastern Austria and the northwestern Balkans [3], showing adaptation to warmer climates than *A.m. mellifera.* The natural distribution of *A.m. iberiensis* covers the Iberian Peninsula and the Balearic Islands [3,4,5], i.e., the warmest regions in Europe. Despite the dominance of honeybees in public awareness, the vast majority of bees are solitary [6]. The red mason bee (*Osmia bicornis*) is a frequent solitary bee in Europe [7]. Each female is fertile and builds her own nest, where eggs are laid in separate cells, along with pollen and nectar provisions [8]. Higher organisms are permanently associated with microbial aggregations formed by bacteria and fungi [9,10]. Pollinators can contribute considerably to the spread of microorganisms by frequent contact with diverse plant surfaces. The ecological community of commensal, symbiotic and pathogenic microorganisms within a given environment is regarded as a microbiome [11]. Surprisingly, the well-studied core gut microbiome of honeybees comprises few bacterial taxa [12,13]. Different castes and even different individuals can display a very different microbiome composition [14,15,16,17]. We assume a long coevolution between hosts and microbiota, which is favored by the social structure of honeybees. Supportive phylogenetic analyses suggest that the primary intestinal bacterial lineages were acquired concomitantly with eusociality [18]. Moreover, honeybees have a social immune system involving allogrooming and resin collection to defeat microorganisms [19,20]. In solitary bees, the gut microbiome is more diverse and flexible [21,22,23,24]. Here, individuals generally do not interact directly. Nevertheless, the same gut bacteria occur in mother bees, pollen provisions and larvae [25,26]. Intriguingly, the same types of bacteria can be found on flowers [24]. Similar results have been reported for honeybees and other flower-visiting insects [27,28].

Flowers likely represent turnover points for bacterial transfer. A bee touches a flower with its tarsae, antennae or proboscis. The cuticle forms the insect’s outer layer. The exoskeletal microbiome composition likely depends on direct interactions of bees with their environment. Bees collect pollen from flowers and store it on their hind legs (honeybees) or on the hairs on the ventral side of their abdomen (red mason bees). Thus, the exchange of microorganisms between flowers and bees is inevitable. Furthermore, honeybee combs are coated with antibiotic propolis [29], which likely influences the composition of their microbiomes. *A.m. mellifera* and *A.m. iberiensis* are known to use propolis extensively [3,30]. Red mason bees build their nests in wooden tubes, mixing mud and other materials with their mandibles. Several solitary bees additionally use mandibular gland secretions or secretions of symbiotic bacteria instead of propolis to disinfect their nest cells [31,32]. Due to their different ranges, *A. mellifera* subspecies and *O. bicornis* are also exposed to different microbial environments. This raises the question of whether and how the microbiome is determined by the environment.

For the first time, in this study, we compared the microbiome composition of honeybee and red mason bee cuticles. We hypothesized that their cuticular microbiomes would differ due to their differing lifestyles, even though they forage in the same habitats. The cuticular microbiome of honeybees should be more strongly regulated by the social immune system and should harbor lower bacterial and fungal diversity and abundance than those of solitary bees. We further analyzed three different honeybee subspecies originating from different regions in Europe that were artificially maintained in the same location. The subspecies should differ in their cuticular microbiome, as they showed behavioral differences [30] and likely visited different flowers due to differences in preferences for floral resources. All three honeybee subspecies should have a less complex cuticular microbiome compared to solitary bees.

## 2. Materials and Methods

### 2.1. Animals

In this study, we analyzed the microbiomes of *A.m. carnica*, *A.m. mellifera*, *A.m. iberiensis* and *O. bicornis* (each *n* = 20). Honeybee in-hive workers were collected from a single colony per subspecies, as described earlier [30]. *O. bicornis* were purchased commercially but were allowed to nest and forage in the same habitat as our honeybee colonies. All bees were maintained in the same habitat. All individuals were flash-frozen in liquid nitrogen and stored at −20 °C.

### 2.2. Cuticle Preparation

Antennae, legs and wings were removed under frozen conditions and used for subsequent analyses. The inner organs were carefully removed. The antennae, legs, wings, head capsule, thorax and abdomen without internal organs were summarized as cuticles. In total, five independent replicates per species/subspecies pool were analyzed. Each pool consisted of four individuals.

### 2.3. Nucleic Acid Extraction

Genomic DNA (gDNA) extraction was performed using a Quick-DNA™ Fecal/Soil Microbe Microprep Kit according to the manufacturer’s protocol. The final extraction volume was 40 µL per sample. The nucleic acid concentration was determined using a Qubit™ fluorometer. All extracts were stored at −20 °C.

### 2.4. Quantitative PCR

Bacterial and fungal gene copy numbers were quantified using quantitative PCR (qPCR). The bacterial 16S rRNA gene (16S) was amplified using 341F and 785R primers [31]. Fungal internal transcribed spacer 2 region (ITS) was analyzed with the fITS7 and ITS4 primers [32]. Each reaction contained iTaq Universal SYBR Green Supermix (1×, Bio-Rad) (Hercules, CA, USA), forward/reverse primer (each 300 nM), PCR Enhancer (1×, biotechrabbit) (Berlin, Germany) and the template DNA (1 µL) in a total volume of 20 µL. Each sample, standard or negative control (nuclease-free master mix blank) was analyzed in independent triplicates in 96-well plates. Runs were performed on a C1000 Touch thermocycler combined with a CFX96 Touch Real-Time PCR Detection System (Bio-Rad). Each run consisted of 3 min at 95 °C followed by 40 cycles of 5 s at 95 °C, 30 s at 52/52.7 °C (16S/ITS) and 30 s at 60 °C. Runs were completed by a melting curve analysis (65–95 °C with an increment of 0.5 °C/5 s). The gene copy number was calculated by comparing cycle threshold values (CT) to a standard curve of 10-fold dilutions of gDNA extracted from a known concentration of *Escherichia coli* K-12 (DSM 423) and *Fusarium solani* (DSM 1164) as described above. The standard gDNA concentrations ranged from 5 × 1012 to 5 × 103 (*E. coli*) and from 5 × 1011 to 5 × 103 gene copies (*F. solani*). Multiple dilutions were run simultaneously to check for qPCR inhibition, and 10-fold-diluted DNA extracts were found to be best-suited for qPCR analyses. The CT and qPCR efficiency were calculated by Bio-Rad software CFX manager version 3.1.

### 2.5. Next-Generation Sequencing

We used 25 µL per gDNA extract to generate 16S/ITS amplicons with the primers as described above. Each sample was amplified five times and pooled afterwards for each species to reduce PCR biases. Quality measurements, library preparation and Illumina© sequencing (300 nt paired-end) on a Miseq V3 (Illumina) (San Diego, CA, USA) were performed by LGC Genomics GmbH (Berlin, Deutschland). Raw data were demultiplexed using bcl2fastq (2.17.1.14, Illumina) and sorted by reads and amplicon inline barcodes. Barcode sequences and adapters were removed, and reads shorter than 100 bp were discarded. Primer detection and separation were performed with three mismatches allowed (per primer). Sequences with incomplete primer pairs were discarded. Complementary reads were combined using BBMerge (v34.48, [33]). Operational taxonomic unit (OTU) identification was carried out using Mothur (v1.35.1, [34]). Sequences with ambiguous bases, homopolymeric segments and average quality scores below 33 were removed, and short reads (truncated and non-specific amplicons) were excluded [35]. Chimeras were eliminated by the UCHIME algorithm [36]. Sequences were aligned to the 16S Mothur-Silva SEED r119 reference [37]. OTUs were selected by clustering with an identity level of 97% using the average neighbor method and taxonomically classified using the Silva reference classification [38]. ITS2 sequences were clustered using CD-HIT-EST (v4.6.1, [39]) with an identity level of 97% using the most abundant sequence. Taxonomic classification was achieved using the UNITE v6 reference database [40]. Singleton OTUs were discarded. Finally, OTU count tables were created (BIOM format, [41]) for each bacterial OTU and fungal OTU. Ecological and metabolic functions of detected fungal OTUs were predicted using the FungalTraits database [42]. Bacterial OTU functions were predicted using the FAPROTAX v.1.1 database [43]. The functions of each prokaryotic taxa were annotated according to the literature on the characterized strains.

### 2.6. Statistics

Statistical analyses were performed using R (4.1.2). Datasets were initially tested for normal distribution and variance homogeneity (Jarque–Bera test and Levene’s test, respectively). For an OTU composition overview of the bacterial and fungal communities, the relative abundances (A_R_) per sample were calculated: n_i_/N (n_i_ = amount sequence reads per OTU, N = sum of all bacterial/fungal reads). Visualization of A_R_ was conducted with OriginPro 2022 (OriginLab Corporation, Northampton, MA, USA). Estimation of alpha diversity was performed using the R package ‘vegan’ (2.5.7). Detrended correspondence analysis (DCA) and non-metric multidimensional scaling (NMDS) were used to calculate beta diversity using R packages ‘FactoMineR’ (2.4) and ‘vegan’. Statistical differences between the species/subspecies were analyzed using PERMANOVA and ANOSIM analysis with the ‘vegan’ package. To test for significant differences in bacterial and fungal gene copy numbers or different diversity indices, we applied linear regression analysis with subspecies nested in species. The ‘emmeans’ package (1.7.2) was used to compute contrasts using Tukey’s range test. Data visualization was performed with R packages ‘ggplot2’ (3.3.3), ‘ggpubr’ (0.4.0), ‘cowplot’ (1.1.1), ‘magick’ (2.7.0) and ‘png’ (0.1–7). The bacterial and fungal genera that differed significantly among the bee species/subspecies were identified using indicator species analysis conducted using the “multipatt” function in the ‘indicspecies’ package, which calculates indicator values with the ‘r.g’ function.

## 3. Results

### 3.1. Bacterial and Fungal Gene Copy Numbers

The cuticular microbiome of *A. mellifera* revealed significantly lower bacterial and fungal gene copy numbers than those of *O. bicornis* (Figure 1). Bacterial gene copy numbers did not differ between honeybee subspecies (Figure 1A). Fungal gene copy numbers were significantly lower in *A.m. mellifera* than *A.m. iberiensis* (Figure 1B).

### 3.2. Cuticular Microbiome Composition

The microbiomes of *O. bicornis* and *A. mellifera* differed significantly in their compositions (Figure 2 and Appendix A). The most abundant bacterial genera of the honeybees’ cuticular microbiome were *Lactobacillus*, *Bifidobacterium*, *Gilliamella*, *Rhizobiales*, *Saccharibacter* and *Snodgrassela* (relative abundance > 5%). In contrast, *Spiroplasma* was the most abundant taxon and also an indicator genus for the *O. bicornis* cuticle (Figure 2A; Table 1). *Lactobacillus* and *Citrobacter* were the next most abundant bacterial genera.

The cuticular fungal microbiome of *O. bicornis* differed significantly from that of *A. mellifera* (Figure 2B), with *Bettsia* (indicator species, Table 1), *Ascomycota* (non-classified), *Capnoidales* and *Monascus* as the most abundant genera (relative abundance > 5%). Additionally, a significant difference between the *A.m. mellifera* and *A.m. iberiensis* fungal microbiome was observed (*p* = 0.0391). The most abundant genera on the cuticular fungal microbiome of *A. mellifera* bees were *Alternaria*, *Capnoidales* (non-classified), *Metschnikowia*, *Saccharomyces*, *Melampsora* and *Aurebasidium*.

### 3.3. Cuticular Microbiome Diversity

The cuticular bacterial and fungal genera richness was highest in *O. bicornis* and *A.m. iberiensis*, while the other *A. mellifera* subspecies had a significantly lower richness (Figure 3A). Overall, 27 bacterial and 24 fungal genera were detected on all bees (Appendix A). Eighty-one bacterial genera were exclusive to *O. bicornis*. Bacterial genera in the *A. mellifera* subspecies were considerably lower. Each subspecies has exclusive bacterial genera (*A.m. iberiensis*, 75; *A.m. carnica*, 10; *A.m. mellifera*, 15). Similarly, *O. bicornis* had the highest number of exclusive fungal genera (108), while *A. mellifera* showed many fewer exclusive fungal genera (*A.m. carnica*, 7; *A.m. mellifera*, 12; *A.m. iberiensis*, 30).

The Shannon index of the cuticular microbiomes was similar between all bees. Differences in richness, evenness, Chao1 and ACE were found between *A. mellifera* and *O. bicornis* (Table 2). Within the honeybee subspecies, *A.m. iberiensis* and *A.m. mellifera* differed significantly only in richness, Chao1 and ACE (Table 2). In summary, the cuticular microbiome diversity of *A.m. iberiensis* was more similar to that of *O. bicornis* than to the other two honeybee subspecies (Figure 3). The cuticular bacterial and fungal community compositions were very similar between the three honeybee subspecies but dissimilar to those of *O. bicornis* (Figure 4 and Appendix A).

The composition pattern of cuticular bacterial functions significantly differed between *O. bicornis* and both *A.m. carnica* (*p* = 0.0262) and *A.m. iberiensis* (*p* = 0.0332) but did not differ from that of *A.m. mellifera* (*p* = 0.0619). *O. bicornis* showed the highest diversity in bacterial functions. Nevertheless, all four groups were dominated by bacterial communities capable of nitrogen fixation and nitrate reduction (Figure 5A). *O. bicornis* also differed from honeybees in the composition of fungal functions (*A.m. mellifera*, *p* = 0.0079; *A.m. carnica*, *p* = 0.009; *A.m. iberiensis*, *p* = 0.0082). The high proportion of animal parasites is noteworthy. While the cuticles of honeybees were dominated by plant pathogenic fungi compared to *O. bicornis*, *A.m. iberiensis* differed significantly from *A.m. mellifera* (*p* = 0.0375) in its fungal functional community composition but not from *A.m. carnica* (*p* = 0.1847).

In total, fifteen indicator species were identified that were associated with the respective bee species. The most bacterial indicator species were found for *O. bicornis*, followed by *A.m. iberiensis* (Table 1). In total, 51 fungal indicator species were found in all bee species, while *Trimmatostroma* was a fungal indicator species for both *O. bicornis* and *A. m iberiensis* (Table 1). Similar to the bacterial indicator species, *O. bicornis* also has the most fungal indicator species (38 genera), followed by *A.m. iberiensis* (8), *A.m. mellifera* (4) and *A.m. carnica* (1).

## 4. Discussion

### 4.1. Social Bees Display a Reduced Cuticular Microbiome

We hypothesized that microorganisms on the cuticle of *O. bicornis* would be more abundant and more diverse than those of *A. mellifera*, as the latter has an effective social immune system [19,20]. Indeed, we found significant differences in the cuticular microbiome between the two species. The microbial communities of red mason bees were more diverse (Figure 2 and Figure 4). For example, *Lactobacillus*, which is typical of the honeybee gut [44], was less abundant on the cuticle of red mason bees (Figure 2). The microbiome of solitary bees is considered to be shaped by the environment rather than by intraspecific transmission [45]. Even *Lactobacillus* members present on *O. bicornis* could be obtained from flowers [23]. We demonstrated, for the first time, a high abundance of the genus *Spiroplasma* on *O. bicornis*, which is potentially pathogenic [46]. These differences probably have little functional significance. In both bee species, chemoheterotrophy and fermentation are the predominant bacterial functions. This can be explained by their lifestyles. Honeybees have few environmental contacts, consisting mainly of their hive and the plants they visit. This also applies to red mason bees. However, honeybees secrete wax for nest building [47]. *Osmia* bees need to collect soil material [48] and are likely to be exposed to multiple soil microbes. Soil comprises a wide range microbial habitats [49], which may account for additional functional communities (e.g. nitrogen fixation and nitrate reduction; Figure 5).

Several fungal genera were absent, whereas others seem to be more specific to *O. bicornis* (Figure 2). This also affected the fungal functional composition. In honeybees, plant pathogens appear to be important, while they account for a lower abundance in red mason bees, where animal parasites were more prominent (Figure 5). Such findings are unlikely for honeybees because they use propolis to combat pathogens [50], combined with frequent cleaning and allogrooming [19,51,52].

Sooty mold fungi seem to be omnipresent on honeybee cuticles, despite the low relative abundance. This common plant disease is associated with honeydew, which honeybees collect [53]. These fungi are reduced on *O. bicornis* cuticles. It is unclear whether *O. bicornis* collect honeydew under natural conditions and whether other cuticular microbes can reduce sooty mold, although they have been shown to collect honeydew under semi-field conditions [54]. In addition, we detected twelve nectar-saprotrophic fungal species on honeybees, but only three species occurred in *O. bicornis*, with significantly reduced abundances (Figure 2). Although honeybees avoid yeast-inoculated nectar [55,56], it seems plausible that the processing of the nectar favors certain fungal species [57]. Here, ripe fruit juice collection [58] may be an additional source of infection.

### 4.2. Honeybee Subspecies from the Same Environment Resemble Similar Cuticular Microbiomes

Differences in microbiome quantities and composition of the different honeybee subspecies were moderate (Figure 2 and Figure 4). *A.m. mellifera*’s cuticle harbors some microbial genera that were absent or reduced in the other two subspecies. However, the functional bacterial composition appears similar, with chemoheterotrophy and fermentation dominating (Figure 5). This suggests a negligible maternal effect on the cuticular microbiome, as the environment of the colonies was identical, whereas the queens originated from different European locations. We observed similar results with the fungal cuticular microbiome. Although *A.m. mellifera* was colonized by some additional genera, the functional composition was similar to that of the other two subspecies (Figure 5). Only plant pathogens were less abundant in *A.m. mellifera*. This subspecies is known to use antimicrobiotic propolis extensively [3]. However, this is also true for *A.m. iberiensis* and, therefore, cannot explain these differences. We conclude that environmental microbiomes are more relevant for the cuticular microbiome of bees than the genetic imprint of the subspecies.

### 4.3. The Cuticular Microbiome Differs from the Gut Microbiome

The gut microbiome of adult worker honeybees usually comprises members of up to nine distinct bacterial genera [44,59,60], some of which were also found on the cuticle (Figure 2). Surprisingly, these bacteria are rarely found in the hive [61,62,63]. It is possible that some of these bacteria also perform important cuticle functions. For example, the prominent *Lactobacillus* can contribute to a low pH, which is an important defense mechanism against pathogens [64]. Maybe they also perform this role on the cuticle. However, we can also detect fewer common gut-typical genera on the cuticle. These include, for example, *Bifidobacterium*, *Gilliamella* and *Snodgrassella*, which are thought to be important in fighting certain pathogens [65,66,67,68,69]. Since horizontal transmission is likely not restricted to the gut [70], allogrooming forces constant repopulation. Other genera, such as *Frischella* and *Bartonella*, were not apparent on honeybee cuticles. Here, the conditions are probably not optimal for all gut microbes, which include the aerobic environment, nutrient availability [17], pH [71], temperature conditions [72], increased UV exposure [73] and the microbial community. The fungal microbiome situation is different. We found genera (*Alternaria*, *Aureobasidium* and *Melampsora*) known from the gut [74,75] and the previously unreported *Metschnikowia*. The fungal microbiome of the gut appears to be mainly shaped by the environment and nutrient availability [74,76,77]. This is likely similar for the cuticle, as many of the fungi we found are common in the environment and associated with bees. For instance, *Alternaria* and *Aureobasidium* are ubiquitously distributed [78,79]. Rust genus *Melampsora* is a typical plant pathogen [80]. Furthermore, *Metschnikowia* is likely distributed by pollinators [81,82]. In the future, it will be necessary to analyze which species are bee-specific and colonize the cuticle permanently and which are temporary guests. Another question that needs to be answered is the extent to which the cuticle microbiome, as a whole, as well as certain representatives, actually influences the gut microbiome and, thus, bee health. The gut microbiome has been shown to have a positive effect on food utilization and defense against certain infections [83]. The cuticular microbiome may also play an important role in fending off pathogens, given that the cuticle is the first direct contact with the environment.

## 5. Conclusions

Herein, we compared, for the first time, the cuticular microbiome of the social *A. mellifera* with that of the solitary *O. bicornis*. As expected, honeybees had a less diverse microbiome, most likely due to their social lifestyle. Furthermore, within different honeybee subspecies living in the same habitat, we found only minor differences in their cuticular microbiome, suggesting a minor effect of genotype. In future experiments, it will be exciting to more closely study which particular hygienic behaviors have a specific impact on the cuticular microbiome of honeybees.

## Figures and Tables

**Figure 1 microorganisms-11-02780-f001:**
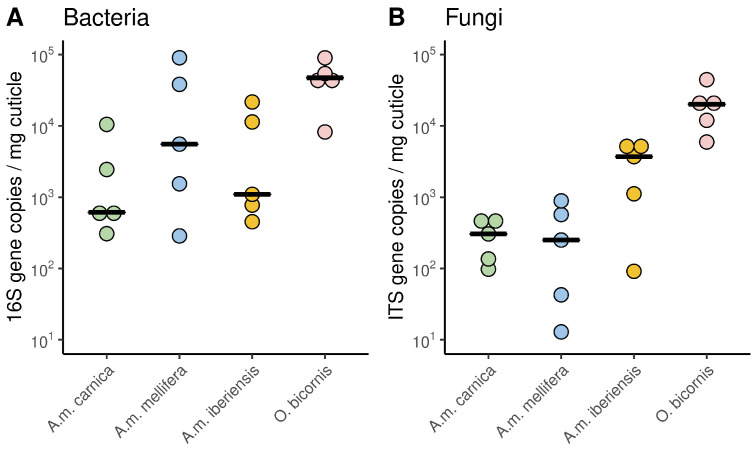
Bacterial and fungal gene copy numbers of the cuticular microbiome from *A. mellifera* and *O. bicornis*. (**A**) Bacterial gene copy numbers differed significantly between species (R^2^ = 0.31, F_(3, 16)_ = 3.885, *p* = 0.029). Significant differences were present between *O. bicornis* and *A. mellifera* (*p* = 0.0075) but not between honeybee subspecies (*p* > 0.05). (**B**) Fungal gene copy numbers differed significantly between species and subspecies (R^2^ = 0.65, FF_(3, 16)_ = 12.78, *p* = 0.0002). Differences not only occurred between *O. bicornis* and *A. mellifera* (*p* = 0.0001) but also between *A.m. mellifera* and *A.m. iberiensis* (*p* = 0.033) but not in the other two combinations (*p* > 0.05). Each species/subspecies is represented by the median of five independent replicates, each consisting of four individuals.

**Figure 2 microorganisms-11-02780-f002:**
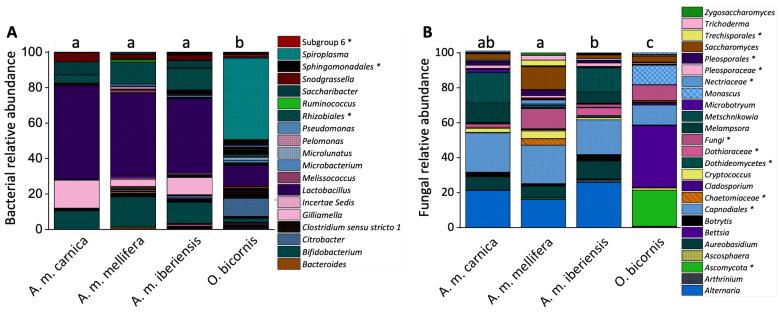
Bacterial and fungal gene copy numbers of the cuticular microbiome from *A. mellifera* and *O. bicornis*. Cuticular bacterial and fungal community composition of *A. mellifera* and *O. bicornis*. Relative sequence read abundances of the bacterial (**A**) and fungal (**B**) communities at the genus level. The legend indicates all genera with a relative sequence read abundance >1%. Each species/subspecies is represented by the mean of five independent replicates, each consisting of four individuals. Different letters indicate statistically significant differences according to one-way non-parametric multivariate analysis (*p* < 0.05). Unclassified members of the respective taxon are marked with *. The full legend is presented in Appendix A.

**Figure 3 microorganisms-11-02780-f003:**
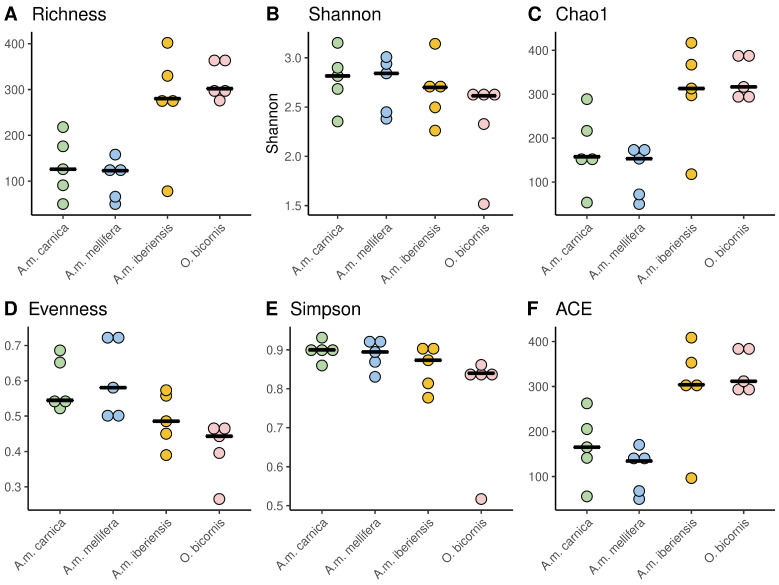
Diversity indices of the cuticular microbiomes of honeybee subspecies and red mason bees. Each species/subspecies is represented by the median of five independent replicates, each consisting of four individuals. Statistical comparisons are summarized in Table 2. Chao1: Chao1 Index; ACE: abundance-based coverage estimator.

**Figure 4 microorganisms-11-02780-f004:**
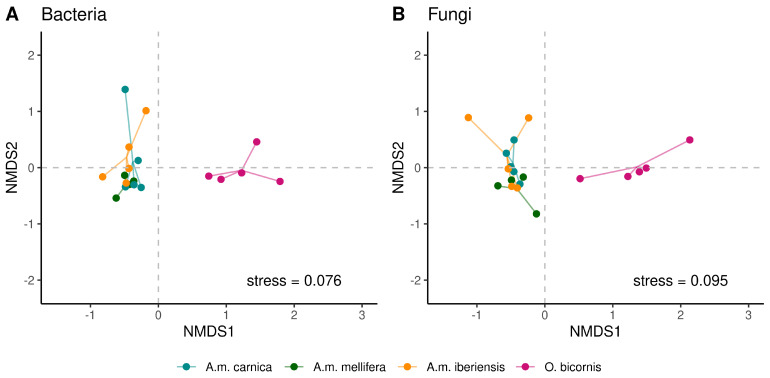
Non-metric multidimensional scaling (NMDS) of the cuticular bacterial (**A**) and fungal (**B**) community composition on *A. mellifera* subspecies and *O. bicornis*. Please see the figure legend for the color code. Each species/subspecies is represented by one of five independent replicates, which each consisted of four individuals. The lines emphasize the centroid of the communities, which represents the mean community composition of a species/subspecies. Data were also supported by DCA analysis (Appendix A).

**Figure 5 microorganisms-11-02780-f005:**
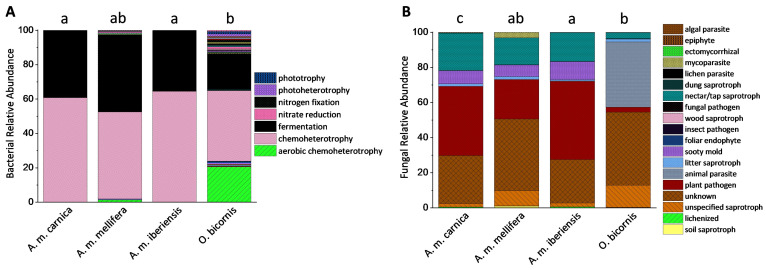
Functional assignment of the cuticular bacterial (**A**) and fungal community composition (**B**) of three honeybee subspecies and red mason bees. Each species/subspecies is represented by the median of five independent replicates, each consisting of four individuals. Different letters indicate statistically significant differences according to one-way non-parametric multivariate analysis (*p* < 0.05). The full legend is presented in Appendix A.

**Table 1 microorganisms-11-02780-t001:** Cuticular bacterial and fungal indicator species analysis of three honeybee subspecies and red mason bees. The stat value and significance of each species are ranked according to the respective stat value.

	Bee Species/Subspecies	Stat	*p*-Value		Family	Genus	Function
Bacteria	*A.m. iberiensis*	0.695	0.0048	**	Lactobacillales uncl.	Lactobacillales uncl.	NA
		0.669	0.0322	*	Elev-16S-1332	Elev-16S-1332 uncl.	NA
		0.665	0.0322	*	Chitinophagaceae	Chitinophagaceae uncl.	aerobic chemoheterotrophy chemoheterotrophy
		0.614	0.0367	*	Rhodospirillales Incertae Sedis	Candidatus *Alysiosphaera*	NA
		0.456	0.0355	*	Solirubrobacteraceae	*Solirubrobacter*	aerobic chemoheterotrophy chemoheterotrophy
		0.423	0.0177	*	Sphingomonadaceae	*Sphingomonas*	aerobic chemoheterotrophy chemoheterotrophy
	*O. bicornis*	0.833	0.0030	**	Xanthomonadales Incertae Sedis	*Steroidobacter*	aerobic chemoheterotrophy
		0.782	0.0003	***	Hyphomicrobiaceae	*Rhodoplanes*	phototrophy
		0.747	0.0003	***	Actinobacteria uncl.	*Actinobacteria* uncl.	NA
		0.744	0.0003	***	Rhodobiaceae	*Rhodobium*	aerobic chemoheterotrophy photoheterotrophy phototrophy
		0.707	0.0303	*	Planctomycetaceae	Pir4 lineage	chemoorganotrophy
		0.701	0.0342	*	Nocardiaceae	*Nocardia*	aerobic chemoheterotrophy chemoheterotrophy
		0.661	0.0003	***	Spiroplasmataceae	*Spiroplasma*	parasitic
		0.657	0.0347	*	Nocardioidaceae	*Kribbella*	chemoorganotrophy
		0.629	0.0303	*	Verrucomicrobiaceae	Verrucomicrobiaceae uncl.	aerobic chemoheterotrophy chemoheterotrophy
Fungi	*A.m. mellifera*	0.659	0.0368	*	Phaeosphaeriaceae	Phaeosphaeriaceae uncl.	litter saprotroph
		0.608	0.0406	*	Tremellales uncl.	*Cryptococcus*	unspecified saprotroph
		0.600	0.0368	*	Eremascaceae	*Eremascus*	unspecified saprotroph
		0.512	0.0083	**	Nectriaceae	Nectriaceae uncl.	plant pathogen animal parasite
	*A.m. iberiensis*	0.736	0.0008	***	Xylariaceae	*Xylaria*	wood saprotroph
		0.682	0.0383	*	Ascomycota uncl.	*Torula*	foliar endophyte
		0.678	0.0383	*	Tremellomycetes uncl.	*Tremellomycetes* uncl.	NA
		0.610	0.0269	*	Pleosporaceae	*Alternaria*	plant pathogen
		0.593	0.0383	*	Xylariales uncl.	*Dinemasporium*	litter saprotroph
		0.526	0.0424	*	Metschnikowiaceae	*Metschnikowia*	nectar tap saprotroph
		0.440	0.0436	*	Herpotrichiellaceae	*Exophiala*	animalparasite
		0.434	0.0438	*	Amphisphaeriaceae	Amphisphaeriaceae uncl.	plant pathogen
	*A.m. carnica*	0.7	0.0087	**	Microbotryaceae	*Microbotryum*	plant pathogen
	*O. bicornis* ^1^	0.900	0.0001	***	Davidiellaceae	*Cladosporium*	litter saprotroph
		0.889	0.0001	***	Polyporaceae	*Fomes*	wood saprotroph
		0.877	0.0001	***	Mycosphaerellaceae	*Mycosphaerellaceae* uncl.	plant pathogen
		0.873	0.0001	***	Capnodiales uncl.	*Toxicocladosporium*	plant pathogen
		0.845	0.0005	***	Capnodiales uncl.	*Pseudotaeniolina*	litter saprotroph
		0.827	0.0037	**	Cystofilobasidiaceae	*Guehomyces*	NA
		0.816	0.0001	***	Saccharomycetaceae	Saccharomycetaceae uncl.	nectar tap saprotroph
		0.805	0.0001	***	Cordycipitaceae	Cordycipitaceae uncl.	animal parasite
		0.802	0.0001	***	Ascosphaeraceae	*Bettsia*	animal parasite
		0.799	0.0003	***	Trichocomaceae	*Aspergillus*	unspecifiedsaprotroph

^1^ excerpt of the 10 genera with the highest Stat-value; Signif. Codes: *** = 0, ** = 0.001, * = 0.01.

**Table 2 microorganisms-11-02780-t002:** Statistical comparison of diversity indices for the cuticular microbiomes of honeybees and red mason bees (see also Figure 3). ns: *p* ≥ 0.05, *: *p*< 0.05, **: *p*< 0.01.

Index		Contrast	*p*-Value	
Richness	R^2^ = 0.58 F_(3,16)_ = 9.696 *p* = 0.0007	*A. mellifera* vs. *O. bicornis* *A.m. carnica* vs. *A.m. mellifera* *A.m. carnica* vs. *A.m. iberiensis* *A.m. mellifera* vs. *A.m. iberiensis*	0.0014 0.8306 0.0249 0.0076	** ns * **
Chao1	R^2^ = 0.52 F_(3,16)_ = 7.849 *p* = 0.0002	*A. mellifera* vs. *O. bicornis* *A.m. carnica* vs. *A.m. mellifera* *A.m. carnica* vs. *A.m. iberiensis* *A.m. mellifera* vs. *A.m. iberiensis*	0.005 0.6236 0.0544 0.0083	** ns ns **
Evenness	R^2^ = 0.42 F_(3,16)_ = 5.514 *p* = 0.0085	*A. mellifera* vs. *O. bicornis* *A.m. carnica* vs. *A.m. mellifera* *A.m. carnica* vs. *A.m. iberiensis* *A.m. mellifera* vs. *A.m. iberiensis*	0.0036 0.9508 0.2180 0.1318	** ns ns ns
ACE	R^2^ = 0.55 F_(3,16)_ = 8.634 *p* = 0.0012	*A. mellifera* vs. *O. bicornis* *A.m. carnica* vs. *A.m. mellifera* *A.m. carnica* vs. *A.m. iberiensis* *A.m. mellifera* vs. *A.m. iberiensis*	0.003 0.5574 0.0538 0.0065	** ns ns **

## Data Availability

Our dataset is publicly available via NCBI accession number PRJNA879967: https://www.ncbi.nlm.nih.gov/bioproject/PRJNA879967/.

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
