# Peer review of "Solitary Bees Host More Bacteria and Fungi on Their Cuticle than Social Bees"

_microorganisms, 2023, doi:10.3390/microorganisms11112780_

Round 1

Reviewer 1 Report

Comments and Suggestions for Authors

Major revision

1.      This study used four individuals as a pool, the amount is not enough.

2.      In figure 3, the author described each species/subspecies is represented by the median of five independent replicates, each consisting of 5 individuals.

3.      The part of discussion might describe the difference in microbiome for function on pathogens resistance in more detail.

Minor revision

1. The author described that O. bicornis were purchased commercially, and could you provide more information because the O. bicornis might be difference in composition of cuticular microbiome between commercial and wild colony.

2. line 20, [] is not reference number.

Author Response

Response letter

We are very grateful to the reviewers for their helpful comments. We answered all comments/questions and additionally provide the line number to track the changes in the manuscript. We provide two versions of the revised manuscript. One where all the changes are highlighted in magenta and another where the changes are not highlighted.

Reviewer 1:

Major revision

  1. This study used four individuals as a pool, the amount is not enough.

Response: Here, the reviewer has probably misunderstood something. Each data point we show consists of four individuals. Thus, in total, we have analysed 20 individuals per species/subspecies. This is described in lines 75-76 and lines 85-86. However, there was unfortunately an error in the figure captions, which we have corrected accordingly (Figures 1, 2, 3,               4, 5).

  1. In figure 3, the author described each species/subspecies is represented by the median of five independent replicates, each consisting of 5 individuals.

Response: Please see our reply to the first comment.

  1. The part of discussion might describe the difference in microbiome for function on pathogens resistance in more detail.

Response: We added some details to the discussion section (lines 264-267, 280-286).

Minor revision

  1. The author described that O. bicornis were purchased commercially, and could you provide more information because the O. bicornis might be difference in composition of cuticular microbiome between commercial and wild colony.

Response:  Yes, they were purchased commercially. However, we assume that our O. bicornis reflect a natural microbiome, because they were allowed to nest and forage in the same natural habitat as our honeybees. We added this information (line 78).

  1. line 20, [] is not reference number.

Response: We added the missing information (line 20).

Reviewer 2 Report

Comments and Suggestions for Authors

The study aims to investigate the cuticular microbiomes of two distinct bee species, the Western honeybee (Apis mellifera) and the solitary red mason bee (Osmia bicornis). Using high-throughput amplicon sequencing techniques, the study analyzes the microbial communities on the bees' body surfaces. This approach offers valuable insights into the composition and complexity of these microbiomes, shedding light on the differences between social and solitary bee species, and can inform more effective conservation and management strategies for bee populations. Furthermore, the consideration of multiple A. mellifera subspecies and the correlation of microbiome variations with climatic adaptation enhance the study's robustness, contributing to a more comprehensive understanding of the factors influencing bee microbiota.

The study exhibits meticulous planning and execution, and the manuscript is well-crafted. Identifying weaknesses in the study proves to be a challenging task. However, I would like to underscore a few aspects that could potentially be improved.

There are numerous multiple comparisons conducted among the various groups in the study. It is worth considering whether Bonferroni (or similar) corrections should be applied to address the issue of inflated Type I error rates associated with these comparisons.

L7: gene based -> gene-based

L20: provide citations, there is empty "[]"

L22-23: you apparently mean A. m. mellifera, not A. mellifera in this place (subspecies, not species as you wrote)

L24: Apis mellifera carnica – in the first occurrence, use the full name, then shorten to A. m. carnica. The same applies to other taxa.

L75-79: The sampling protocol's clarity is somewhat lacking, particularly regarding whether the honeybee individuals were sourced from single or multiple colonies.

Author Response

Response letter

We are very grateful to the reviewers for their helpful comments. We answered all comments/questions and additionally provide the line number to track the changes in the manuscript. We provide two versions of the revised manuscript. One where all the changes are highlighted in magenta and another where the changes are not highlighted.

Reviewer 2:

The study aims to investigate the cuticular microbiomes of two distinct bee species, the Western honeybee (Apis mellifera) and the solitary red mason bee (Osmia bicornis). Using high-throughput amplicon sequencing techniques, the study analyzes the microbial communities on the bees' body surfaces. This approach offers valuable insights into the composition and complexity of these microbiomes, shedding light on the differences between social and solitary bee species, and can inform more effective conservation and management strategies for bee populations. Furthermore, the consideration of multiple A. mellifera subspecies and the correlation of microbiome variations with climatic adaptation enhance the study's robustness, contributing to a more comprehensive understanding of the factors influencing bee microbiota.

The study exhibits meticulous planning and execution, and the manuscript is well-crafted. Identifying weaknesses in the study proves to be a challenging task. However, I would like to underscore a few aspects that could potentially be improved.

There are numerous multiple comparisons conducted among the various groups in the study. It is worth considering whether Bonferroni (or similar) corrections should be applied to address the issue of inflated Type I error rates associated with these comparisons.

Response: P value adjustment was achieved via Tukey’s range test method which is implemented in the R package „emmeans“. We added this information (lines 148-149).

L7: gene based -> gene-based

Response: changed (line 7)

L20: provide citations, there is empty "[]"

Response: We are sorry for this mistake and added the missing information (line 20)

L22-23: you apparently mean A. m. mellifera, not A. mellifera in this place (subspecies, not species as you wrote)

Response: We are grateful for the hint and have changed the text accordingly (line 23).

L24: Apis mellifera carnica – in the first occurrence, use the full name, then shorten to A. m. carnica. The same applies to other taxa.

              Response: changed (line 24, Figure 1, Figure 2)

L75-79: The sampling protocol's clarity is somewhat lacking, particularly regarding whether the honeybee individuals were sourced from single or multiple colonies.

Response: Honeybee individuals were sourced from single colonies per subspecies. We added this information (line 76-77).